# Comparative Genomics Revealing Insights into Niche Separation of the Genus *Methylophilus*

**DOI:** 10.3390/microorganisms9081577

**Published:** 2021-07-24

**Authors:** Nana Lin, Ye Tao, Peixin Gao, Yan Xu, Peng Xing

**Affiliations:** 1School of Civil Engineering, Southeast University, Nanjing 210096, China; 220191214@seu.edu.cn (N.L.); xuxucalmm@seu.edu.cn (Y.X.); 2State Key Laboratory of Lake Science and Environment, Nanjing Institute of Geography and Limnology, Chinese Academy of Sciences, Nanjing 210008, China; taoye@biozeron.com (Y.T.); gao_px@sina.com (P.G.); 3College of Resources and Environment, University of Chinese Academy of Sciences, Beijing 100049, China

**Keywords:** *Methylophilus*, whole-genome sequencing, comparative genome analysis, pan-genome analysis, orthology analysis, nitrogen metabolism, sulfur metabolism

## Abstract

The genus *Methylophilus* uses methanol as a carbon and energy source, which is widely distributed in terrestrial, freshwater and marine ecosystems. Here, three strains (13, 14 and QUAN) related to the genus *Methylophilus*, were newly isolated from Lake Fuxian sediments. The draft genomes of strains 13, 14 and QUAN were 3.11 Mb, 3.02 Mb, 3.15 Mb with a G+C content of 51.13, 50.48 and 50.33%, respectively. ANI values between strains 13 and 14, 13 and QUAN, and 14 and QUAN were 81.09, 81.06 and 91.46%, respectively. Pan-genome and core-genome included 3994 and 1559 genes across 18 *Methylophilus* genomes, respectively. Phylogenetic analysis based on 1035 single-copy genes and 16S rRNA genes revealed two clades, one containing strains isolated from aquatic and the other from the leaf surface. Twenty-three aquatic-specific genes, such as 2OG/Fe(II) oxygenase and diguanylate cyclase, reflected the strategy to survive in oxygen-limited water and sediment. Accordingly, 159 genes were identified specific to leaf association. Besides niche separation, *Methylophilus* could utilize the combination of ANRA and DNRA to convert nitrate to ammonia and reduce sulfate to sulfur according to the complete sulfur metabolic pathway. Genes encoding the cytochrome *c* protein and riboflavin were detected in *Methylophilus* genomes, which directly or indirectly participate in electron transfer.

## 1. Introduction

Methylotrophic bacteria are characterized by their use of reduced carbon substrates (methane, methanol and other methylated compounds) as their sole sources of carbon and energy [1]. These bacteria are divided into facultative (microbes that can use both C1 and multicarbon compounds as substrates) and obligate (microbes that can only use C1 substrates) methylotrophs [1]. From a phylogenetic point of view, they are affiliated with Alpha-, Beta- and Gammaproteobacteria, Actinobacteria and Verrucomicrobia, respectively [2]. In nature, methylotrophic bacteria are widely distributed in terrestrial habitats (such as wastewater treatment plants [3], wetlands [4], municipal biowaste [5]), freshwater and marine ecosystems, which participate in carbon and nitrogen cycling [6]. As a carbon source, methanol is unique in that it can only be metabolized by methylotrophic bacteria, such as members of the family Methylophilaceae and Methylobacteriaceae [7]. Bioconversion drives sustainable and clean utilization of methanol. Methylotrophic bacteria can be used as hosts to construct cell factories and synthesize amino acids and cell proteins through the abundant one-carbon compounds in nature [8]. In addition, metabolic engineering and synthetic biology strategies are adopted to improve methanol utilization efficiency and promote methanol bioconversion for the Methylotrophs. Thus, it is of great practical significance to study the metabolism of methylotrophic bacteria [9].

The genus *Methylophilus* comprises non-halophilic, obligately and restricted facultatively methylotrophic bacteria [10] belonging to the family Methylophilaceae, order Nitrosomonadales, class Betaproteobacteria and phylum Proteobacteria. Thus far, eight species (*Methylophilus aquaticus*, *M. flavus*, *M. glucosoxydans*, *M. leisingeri*, *M. luteus*, *M. methylotrophus*, *M. quaylei* and *M. rhizosphaerae*) are nominated in the genus [10]. They use methanol as a carbon and energy source and oxidize methanol by the catalyst methanol dehydrogenase (MDH), the activity of which is regulated by the type of carbon source [11]. In recent years, worldwide attention has been paid to nonmethanotrophic methanol users because of the increase in methanol emissions (82–273 Tg/year) [6]. Thus the insight into the genus *Methylophilus* can provide solutions for sustainable utilization of methanol and its industrial application. We aimed to expand the cultivation of this functional group and improve our understanding of its role in lake ecosystems.

*Methylophilus* species were detected in hyperoxia, hypoxia and anoxia environments, and especially at the interface of hyperoxia–hypoxia [12], indicating that they could survive under oxygen-limited conditions as aerobic bacteria traditionally. Moreover, colonization of *Methylophilus* spp. in different habitats is the result of mutual adaptation between bacteria and the environment. We expect to enhance the understanding of the survival mechanism of the genus *Methylophilus* under oxygen-limited conditions. Thus, in this study, three newly isolated methylotrophic strains (*Methylophilus* sp. 13, *Methylophilus* sp. 14 and *Methylophilus* sp. QUAN) were obtained from an enriched culture of Lake Fuxian sediment samples. We speculated that *Methylophilus* strains isolated from various habitats could harbor differences in their functional gene numbers and categories. 

In recent years, with the rapid development of high-throughput sequencing technology, massive amounts of microbial genomic data revealed the possibility of studying various microbes. The advantages of comparative genomic analysis over the conventional approach lie in the ability to guide the discovery of functional genes structure, predict the unique metabolic pathways, and obtain a functional potential for the sequenced microbes. A number of genome sequences available in the public database also provided the opportunity to go deep into the genus *Methylophilus*. The comparative genomic analysis was performed using three newly isolated *Methylophilus* strains and the other 15 strains belonging to the genus *Methylophilus* that had been sequenced and archived in the database to understand genome-level differences and similarities among them. Pan- and core-genome analyses were performed based on average nucleotide identity (ANI) and average amino acid identity (AAI) as well as orthologic, phylogenetic, metabolic and statistical analyses. The comparative genomic analysis could help us better understand the impact of the heterogeneous environment on the genus *Methylophilus* at the genomic level.

## 2. Materials and Methods

### 2.1. Strain Isolation and DNA Extraction

The samples were taken from the sediment column of Lake Fuxian in Yunnan Province and the enrichment culture of methylotrophs was grown in a liquid medium (NMS), containing (in grams per liter of distilled water) KNO_3_, 1.0; KH_2_PO_4_, 0.35; Na_2_HPO_4_·12H_2_O, 0.65; NaHCO_3_, 8.4; Na_2_CO_3_, 3.0; NaCl, 7.5; MgSO_4_·7H_2_O, 0.2; CaCl_2_, 0.02, with the addition of trace elements in the solution, containing (in grams per liter of distilled water) Na_2_EDTA 0.5; FeSO_4_·7H_2_O 0.2; H_3_BO_3_ 0.03; ZnSO_4_·7H_2_O 0.01; MnCl_2_·4H_2_O 0.003; CaCl_2_·6H_2_O 0.02; CuSO_4_·5H_2_O 0.03; NiCl_2_·6H_2_O 0.002; Na_2_MO_4_·2H_2_O 0.003. A set of 100 mL bottles was filled with 20 mL NMS medium, 5.0 g wet sediment and 10 mL CH_4_ (10%, *v*/*v*) in the headspace. We obtained a number of methanotrophic consortia through long-term incubation at 18 °C in the laboratory, which mainly consisted of methanotrophs associated with methylotrophs and a few heterotrophic bacteria. The inoculum was routinely transferred once a week. After 10 generations, the culture was taken out from the liquid medium, diluted to about 10^−3^–10^−6^ and isolated by sieving on NMS agar media containing (in grams per liter of distilled water) KH_2_PO_4_, 2.0; (NH_4_)_2_SO_4_, 2.0; MgSO_4_·7H_2_O, 0.025; NaCl, 0.5; FeSO_4_·7H_2_O, 0.02, agar, 1.5%~2.0% (*w*/*v*), with the addition of 0.5% methanol (*v*/*v*) as a carbon source, which was evenly coated on the surface of agar plates by coating rods after the solidification of NMS agar plates, and inoculated bacteria until methanol was absorbed. Normally after 3–4 times of purification on agar plates, pure *Methylophilus* colonies could be obtained. By following these steps, three strains *Methylophilus* sp. 13, *Methylophilus* sp. 14 and *Methylophilus* sp. QUAN were obtained from the enrichment consortia. 

Genomic DNA was extracted from the cell pellets using a Bacterial DNA Kit (D3350-01, OMEGA) according to the manufacturer’s instructions, and quality control was subsequently carried out on the purified DNA samples. Genomic DNA was quantified by using a Qubit3.0 fluorometer (Thermofisher, Waltham, MA USA). A highly qualified DNA sample (OD260/280 = 1.8~2.0, >3 μg) was used in further sequencing.

The full-length 16S rRNA gene of the three strains was amplified by PCR with universal primers (27F 5′-AGAGTTTGATCCTGGCTCAG-3′; 1429R 5’-GGTTACCTTGTTACGACTT-3′) and sequenced by the Tsingke Biology and Technology Company. The 25 μL reaction mixture contained 0.5 μL of forward primer (10 μM) and 0.5 μL of reverse primer (10 μM), 23 μL of T3 Mix enzyme and 1 μL of DNA templates. The cycling conditions were 10 min at 95 °C for pre-denaturation, followed by 35 cycles of 50 s at 95 °C, 50 s at 52 °C for, 90 s at 72 °C, and a final 10 min at 72 °C for extension. The DNA sequence data were analyzed using BLAST software (http://blast.ncbi.nlm.nih.gov/Blast.cgi, 12 July 2019). The phylogenetic position for each strain was determined by comparison with related taxa (11 *Methylophilus* isolates, 8 *Methylophilus* type species and 2 *Methylotenera* strains) obtained from the GenBank database (http://www.ncbi.nlm.nih.gov/, 10 March 2020). A phylogenetic tree was constructed using the neighbor-joining [13] method in MEGAX [14] (version 10.1.8). According to 16S rRNA gene sequence similarity and phylogenetic analysis, 13, 14 and QUAN were classified as belonging to the genus *Methylophilus.*

### 2.2. Genome Sequencing and Assembly

At least 3 μg of genomic DNA were used for library sequencing construction. Paired-end libraries with insert sizes of ~450 bp were prepared following Illumina’s standard genomic DNA library preparation procedure. Briefly, purified genomic DNA was sheared into smaller fragments of a desired size by Covaris, and blunt ends were generated by using T4 DNA polymerase. After adding an “A” base to the 3′ end of the blunt phosphorylated DNA fragments, adapters were ligated to the ends of the DNA fragments. The desired fragments were purified through gel electrophoresis, then selectively enriched and amplified by PCR. The index tag was introduced into the adapter at the PCR stage as needed, and a library-quality test was done. Finally, the qualified Illumina pair-end library was used for Illumina Miseq sequencing (300bp × 2, Shanghai BIOZERON Co., Ltd.). The raw paired-end reads were trimmed and quality controlled by Trimmomatic with parameters (Truseq PE adaptors, SLIDINGWINDOW:4:15 MINLEN:75, version 0.36 http://www.usadellab.org/cms/uploads/supplementary/Trimmomatic, 7 August 2020). A SPAdes software package was introduced to conduct genome assembly with default parameters [15]. Contigs of fewer than 500 bp were discarded in our study by custom perl scripts. The genomic sequences of three strains (13, 14 and QUAN) were submitted to the GenBank database under accession numbers GCA_015354335.1, GCA_015354345.1 and GCA_015354445.1, respectively.

### 2.3. Gene Prediction and Annotation

Bacterial gene models were identified using GeneMark (Georgia Institute of Technology, Atlanta, GA, USA) [16]. Then, all gene models were compared using a BLAST against non-redundancy in the NCBI, KEGG (http://www.genome.jp/kegg/, 8 August 2020) and COG (http://www.ncbi.nlm.nih.gov/COG, 8 August 2020) databases so that functional annotation by the BLASTp module could be conducted [17]. In addition, tRNA were identified using the tRNAscan-SE (Department of Biomolecular Engineering, University of California Santa Cruz, CA 95064, USA, v1.23, http://lowelab.ucsc.edu/tRNAscan-SE, 8 August 2020) and rRNA were determined using the RNAmmer (Centre for Molecular Biology and Neuroscience and Institute of Medical Microbiology, University of Oslo, NO-0027 Oslo, Norway, v1.2, http://www.cbs.dtu.dk/services/RNAmmer/, 8 August 2020).

### 2.4. Genome Selection and Phylogenomic Analysis

The available genomic sequences were downloaded from the NCBI RefSeq database by using “*Methylophilus*” as the keyword. Among a total of 17 online genomes (until 1 September 2020), GCA_006363915.1 and GCA_006363935.1 represented the same species *Methylophilus medardicus,* and their sequences were completely identical, so GCA_006363935.1 was discarded in the following analysis. To eliminate any possible errors in further comparative genomic analysis, gene models and other genomic elements like tRNA and rRNA were re-identified by the pipeline described above. We also introduced CheckM software to evaluate the quality of all the genomes in this study [18]. Those with a completeness of less than 75% or a contamination larger than 5% were discarded. Therefore, GCA_008015785.1 was also discarded, and the remaining 15 public genomes were analyzed alongside the three genomes obtained in this study (Table 1).

OrthoFinder2 was used to perform a DIAMOND-based all-versus-all gene search on amino acid levels and identified clusters of orthologous genes (OGs) [19,20]. In all, 1035 single-copy OGs were aligned by MUSCLE and concatenated to construct a phylogenetic tree with FastTree under the maximum-likelihood method with *Methylotenera versatilis* 301^T^ and *Methylotenera mobilis* JLW8^T^ as the outgroup [21,22]. The average nucleotide identity (ANI) was calculated as previously described by FastANI with default parameters [23,24], and average amino acid identities (AAI) were calculated to discriminate among different species and genera. An ANI and AAI heatmap was drawn using the “pheatmap” package in an R environment. 

### 2.5. Metabolic and Statistical Analysis

A metabolic functional gene heatmap and metabolic pathway were constructed using the “pheatmap” package in an R environment and Adobe Illustrator CC 2019, respectively. In addition, histograms were performed to identify the gene number contained in different isolation habitats using GraphPad Prism (version 8.2.1). Ordinary one-way ANOVA was used to analyze significant gene number differences among the different habitats and 95% confidence intervals were given.

## 3. Results

### 3.1. General Genomic Features of the Three New Strains

The genomic features of the genus *Methylophilus* are listed in Table 1. The draft genomic sequences of *Methylophilus* sp. 13, 14 and QUAN consist of 3.11 Mb, 3.02 Mb and 3.15 Mb, with scaffold numbers of 14, 15 and 6, coverage of 590, 377 and 405-fold, respectively. Compared to the new isolates in this study, the quality of the genomes obtained from the metagenomic assembly pipeline was relatively low. For example, genome completeness was 78.45% for GCA_003538435. In total, 41, 40 and 42 tRNA genes were detected, and a total of 2998, 2914 and 3109 genes were identified from *Methylophilus* sp. strains 13, 14 and QUAN, respectively. Moreover, the 16S rRNA gene was assembled in all three genomes with a completeness of 100%.

Phylogenetic analysis was performed based on 16S rRNA genes and the 1035 single-copy genes to reveal the evolutionary relationship among the 18 *Methylophilus* strains. The 16S rRNA gene sequence similarity between *Methylophilus* spp. ranged from 97.17 to 100% (Figure 1), among which similarities between strains 13 and 14 was 99.28%, strains 13 and QUAN was 99.28% and strains 14 and QUAN was 100%. Both *Methylophilus* sp. 14 and QUAN had 99.52% 16S rRNA gene similarity with their closest neighbor *M*. *methylotrophus* NCIMB 10515^T^ (AB193724). In addition, *Methylophilus* sp. 13 has its closest relative *Methylophilus quaylei* MT^T^ (AY772089) with 99.58% 16S rRNA gene similarity.

Three newly isolated strains (*Methylophilus* sp. 13, 14 and QUAN) in this study formed significant new branches in the phylogenomic tree (Figure 2, Appendix A). The ANI and AAI value between *Methylophilus* spp. ranged from 78.66 to 99.99%, 81.07% to 100%, respectively. The ANI and AAI value of strain 14 with the closest type strain *M**. methylotrophus* DSM 46235^T^ were 81.69 and 87.03%, respectively. The ANI and AAI value of strain QUAN with the closest type strain *M**. methylotrophus* DSM 46235^T^ were 81.92 and 86.95%, respectively. All the values were lower than the species threshold corresponding to 95% ANI and 90% AAI. Especially, two strains 14 and QUAN harbored 100% similarity according to the phylogenetic analysis based on 16S rRNA genes, but phylogenetic analysis based on 1,035 single-copy genes and the ANI results (91.46%) indicated they diverged into two species, which might be novel species candidates in the genus *Methylophilus*. Since the genome sequence of *M. quaylei* MT^T^ was not available yet, we did not have enough information to decide whether strain 13 would be a novel species candidate. 

Among the 18 *Methylophilus* genomes, six strains (*Methylophilus* sp. 1, *Methylophilus* sp. 42, *Methylophilus* sp. OH31, *Methylophilus* sp. Q8, *Methylophilus* sp. UBA11725 and *Methylophilus* sp. UBA6697) possibly belong to the same species since the intraspecies level is defined at ≥95% ANI [23]; meanwhile, *Methylophilus* sp. Leaf459, Leaf416 and Leaf414 belong to the same species according to their high ANI values (≥95% ANI). In addition, both ANI and AAI analysis supported the finding that the remaining *Methylophilus* species were differentiated from each other (Figure 2 and Appendix A). 

These 18 *Methylophilus* strains were divided into two groups according to their isolated habitats. *Methylophilus* sp. Leaf459, Leaf416 and Leaf414 were isolated from a terrestrial plant leaf, and the remaining species were isolated from aquatic ecosystems. Furthermore, two subgroups, including lake water and sediment, were identified in the aquatic habitats. In addition, *M Rhizosphaerae* CBMB127^T^ and *M. methylotrophus* DSM 46235^T^ were derived from the rhizosphere soil of rice and activated sludge [25,26], respectively. Apparently, phylogenetic analysis revealed that the strains isolated from the leaf were separated from other related representatives of water and sediment. Notably, the commonness and differences among the strains isolated from various environments required further study.

The GC% of *Methylophilus* genomes ranged from 48.24 to 51.35%, with the lowest value belonging to strain *Methylophilus.* Leaf414, and the highest to *Methylophilus**. rhizosphaerae* strain GCA_900100975.1. A one-way ANOVA for GC% of genomes isolated from the three habitats (leaf, water and sediment) was performed, and gave 95% confidence intervals. The strains isolated from leaf had a significantly lower G+C content (~48.26%) than that of the strains isolated from freshwater or sediment ecosystems (*p* < 0.0001), whereas no difference was identified between water and sediment ecosystems (*p* = 0.3721).

### 3.2. Pan-Genome and Orthology Analysis

The gene number of the *Methylophilus* genomes selected in this study ranged from 2509 to 3109, with the lowest value belonging to “*Methylophilus medardicus”* strain GCA_006363915.1, and the highest to *Methylophilus* sp. QUAN. The total gene number of the pan-genome increased and the number of shared genes (core genome) decreased with the addition of newly sequenced strains, which fit the exponential decaying function [27]. Remarkably, a total of 3994 genes were identified in the pan-genome (Figure 3B). The core genome curve showed that it reached a minimum number of 1559 genes and remained constant (Figure 3C). The singleton gene number ranged from 2 to 362, with the lowest value belonging to *Methylophilus* sp. 42, and the highest to *Methylophilus* sp. QUAN. For the three new isolated strains, *Methylophilus* sp. 13 comprised 326 singleton genes for a total gene number of 2998, whereas the strains 14 and QUAN comprised 202 and 362 singletons, respectively. 

We performed a comparison of leaf-specific and aquatic habitat-specific orthogroups (Figure 4, Appendix A). Specifically, the leaf-specific genomes (*Methylophilus* sp. Leaf459, Leaf416 and Leaf414) encoded 159 unique genes, such as methyltransferase, exodeoxyribonuclease, acyl carrier protein, aldehyde oxidase and xanthine dehydrogenase, etc. Whereas the aquatic habitat-specific genomes encoded 23 unique genes, such as PEP-CTERM sorting domain-containing protein, metal-dependent phosphohydrolase, an efflux RND transporter permease subunit, sigma-70 family RNA polymerase sigma factor, peptidyl-prolyl *cis-trans* isomerase, diguanylate cyclase, and poly(3-hydroxybutyrate) depolymerase, etc. It was found that the number of hypothetical proteins specific to the leaf (89 genes) was much higher than those found in aquatic habitats (8 genes).

We found that the genes that came from water were similar to those from sediment; that is to say, genes isolated from aquatic habitats harbored high similarity. For example, the genomes of strains isolated from water and sediment both encoded the efflux RND transporter permease subunit, which belongs to the resistance-nodulation-cell division (RND) family. The genomes encoding the MgtC family protein were present in the aquatic habitat ecosystem, which is the virulence factor required for growth in a low Mg^2+^ medium and for intramacrophage survival and may be involved in regulating membrane potential by activating Na^+^/K^+^-ATPase. The RNA polymerase sigma factor, cytochrome *c* peroxidase, a DNA-binding protein, and the outer membrane protein TolC were all encoded by aquatic habitat-specific genomes. Besides their high similarity, the water and sediment environments also comprised specific genes (1 and 2 genes, respectively, Appendix A). In summary, the strains isolated from the leaf and aquatic habitats showed differences in functional genes, and the strains isolated from water and sediment had a high similarity on the whole, even if differences existed.

### 3.3. Metabolic Analysis

#### 3.3.1. Methanol Metabolism

In methanol metabolism, genes for the functioning of tetrahydromethanopterin (H_4_MPT) were present (Figure 5A,B). Periplasmatic pyrroloquinoline quinone (PQQ)-dependent methanol dehydrogenase (MDH), which converts methanol to formaldehyde, containing a gene cluster (*mxaA/C/D/G/I/J/K/L*), was found [7]. Formaldehyde can be converted to format through a H_4_MPT-dependent pathway containing a series of processes catalyzed by enzymes, including *fae*, *mtdB*, *mch*, *ftr* and *fwdA/B/C*. Notably, these genes were detected in all genomes, indicating that H_4_MPT was the most significant pathway in the methanol metabolism of *Methylophilus*. Moreover, genomes isolated from the same source tended to have uniform gene clusters, indicating that they had a functional similarity in methanol metabolism. Genes encoding trimethylamine monooxygenase (*tmm*), dimethylamine–trimethylamine dehydrogenase (*dmd–tmd*), methylamine–glutamate N-methyltransferase (*mgsA/B/C*) and methylamine dehydrogenase (*mauA/B*) were found in the sedimentary environment, which provided a way to convert trimethylamine N-oxide to formaldehyde under anaerobic conditions [28] (Figure 5C). Genes encoding *mfnD/F* were distinctly identified in the leaf ecosystem, which catalyzed the processes of producing methanofuran (MFR). Additionally, genes encoding *mfnD* presented significant differences among environments (Figure 5C, Appendix A). In summary, the *Methylophilus* strains use methanol via MDHs, convert trimethylamine N-oxide to formaldehyde under the anaerobic environment, and then metabolize formaldehyde through the H_4_MPT pathway [29].

#### 3.3.2. Carbohydrate-Active Enzymes Analysis

The CAZy (Carbohydrate-Active enZYmes) database provided significant genome analysis by family distribution and associated proteins identified in genomes. The enzyme classes currently covered six modules: GHs (glycoside hydrolases), GTs (glycosyl transferases), PLs (polysaccharide lyases), CEs (carbohydrate esterases), AAs (auxiliary activities) and CBMs (carbohydrate-binding modules). The heatmap was conducted according to the CAZy results (Appendix A), which suggested a greater accumulation of carbohydrate-active enzymes in the leaf habitat. We identified 5 CAZy families specific to the leaf specific genomes, including 3 CEs (CE1/4/10), 1GTs (GT2) and 1 AA (AA12). Only one CAZy family (CE1) was specific to aquatic-habitat genomes (Appendix A). 

#### 3.3.3. Nitrogen Metabolism

The *Methylophilus* genomes harbored genes involved in denitrification (*norB*), assimilatory nitrate reduction (*nasA*) and dissimilatory nitrate reduction (*nirB/D*) as well as genes that encode carbonic anhydrase (*cynT*/*cah*), cyanate lyase (*cynS*), and nitrate/nitrite transport system substrate-binding protein (*NRT*/*nrtA*/*nrtB*/*nrtC*; Figure 6A). Genomes isolated from the same habitats tended to have uniform gene clusters, indicating that they had a functional similarity in nitrogen metabolism (Figure 6B). Genes encoding the nitrate/nitrite transport system (*NRT*/*nrtABC*), which can transport extracellular nitrate into cells, were identified in all *Methylophilus* genomes except *M. rhizosphaerae*. The assimilatory nitrate reductase catalytic subunit A (*nasA*), which participated in reducing nitrate to nitrite, was identified specifically in water and sediment strains (Figure 6C). Genes encoding *nirBD*, which can reduce nitrite to ammonia, were present in strains isolated from all three sources. The above findings suggested that *Methylophilus* might metabolize nitrate by using assimilatory nitrate reduction combined with dissimilatory nitrate reduction to produce ammonia, the essential nitrogen source for growth. Genes encoding nitric oxide reductase (*norB*), which can reduce nitric oxide to nitrous oxide, were only present in strains isolated from leaf and sediment. *Methylophilus* genomes also encoded two carbonic anhydrases (*cah* and *cynT*) and one cyanate lyase (*cynS*), which was involved in a cyanate degradation pathway. Additionally, genes encoding *nasA* and *norB* presented significant differences among environments (Figure 6C, Appendix A).

#### 3.3.4. Sulfur Metabolism

Genes for the function of sulfur metabolism were presented in *Methylophilus* genomes (Figure 7A). The sulfur-oxidizing protein *soxY*, assimilatory sulfate reduction (*Cys*) operon, cytochrome subunit of sulfide dehydrogenase *fccA*, sulfate transport system substrate-binding protein *sbp*, sulfide quinone oxidoreductase *sqr* and dimethylsulfone monooxygenase *sfnG* genes were detected in all three environments, suggesting that *Methylophilus* can synthesize sulfur-containing proteins. Genomes isolated from the same source tended to have uniform gene clusters, indicating that they had a functional similarity in sulfur metabolism (Figure 7B). The sulfide quinone oxidoreductase (*sqr*), which is involved in the oxidation of reduced sulfur, is responsible for oxidizing sulfide (HS^−^) to elemental sulfur (S_0_) [30]. Additionally, *ssuC* gene encoding sulfonate transport system permease protein harbored significant variation among environments (Figure 7C, Appendix A). Therefore, the *Methylophilus* strains can use some sulfur compounds and play a role in the sulfur cycle as the complete sulfur metabolic pathway has been found in the genomes.

#### 3.3.5. Cellular Electron Transfer for Respiration

Studies have shown that the respiration of cellular electron acceptors provides a strategy for the study of microbial survival mechanisms under oxygen-limited conditions [31]. Based on genomic analysis, cytochrome *c* and riboflavin were found to be involved in cellular mineral respiration, which is the basis for cellular electron transfer between minerals and microorganisms [32].

In our study, genes encoding the cytochrome *c* protein were identified in 18 genomes, such as cytochrome *c* oxidase (*ccoP/O/N*, *cox11/15/A/B/C*), cytochrome *c* reductase (*CYC1*, *cytB*) and cytochrome *c* protein (*CYC*) (Figure 8). Additionally, the genes mentioned above (except *CYC*) were identified in all genomes across the three habitats, indicating that *Methylophilus* can mediate cellular electron transfer for respiration via cytochrome *c* proteins. As a common electron shuttle, riboflavin provides the approach to shorten the distance between minerals and microorganisms, and enhances the ability of electron transfer [33]. Additionally, genes encoding riboflavin, which indirectly mediates electron transfer, were detected in the genomes, such as GTP cyclohydrolase II (*ribA*), 3,4-dihydroxy 2-butanone 4-phosphate synthase (*ribBA*), 5-amino-6-(5-phosphoribosylamino) uracil reductase (*ribD*), 6,7-dimethyl-8-ribityllumazine synthase (*ribH*), riboflavin synthase (*ribE*) and riboflavin kinase/FMN adenylyltransferase (*ribF*). The genes mentioned above were found to exist in all three environments, which suggested that *Methylophilus* can provide a potential pathway for riboflavin biosynthesis, thus mediating cellular electron transfer through soluble shuttle proteins. According to the statistical analysis performed via one-way ANOVA (Figure 8, Appendix A), genes encoding cytochrome *c* and riboflavin proteins were not significantly different among habitats except for the *CYC* gene (cytochrome *c*), which presented significant differences between leaf and sediment.

In summary, the genes encoding cytochrome *c* and riboflavin were detected in the genus *Methylophilus*, which provide the blueprint for the study of the cellular electron transfer mechanism. Additionally, this study provided new insights into the metabolic capacity of methylotrophic bacteria and their survival mode in an oxygen-limited environment, and highlighted the significance of studying the cellular electron transfer mechanism for further exploration of their ecological role.

## 4. Discussion

In this study, all the sequenced genomes of *Methylophilus* obtained from the GenBank database received attention and a comparative genomic analysis was conducted from the perspective of the pan-genome. The analysis of newly isolated strains in this study (*Methylophilus* sp. 13, *Methylophilus* sp. 14, *Methylophilus* sp. QUAN) along with other 15 strains belonging to the genus *Methylophilus*, provided insight into the niches separation of *Methylophilus spp*.

### 4.1. Adaptation to Specific Habitats from Genomic View

Genomes obtained from the three environments were significantly different from each other, especially in the encoding of enzymes. The habitat-specific genes encoding relevant enzymes within the genus *Methylophilus* might be the results of the adaptation to their environment. The genus *Methylophilus* could survive hyperoxia, hypoxia and anoxia environments, especially under oxygen-limited conditions [12]. Hydroxylation is an important bacterial regulation mechanism, and the response of hydroxylases to hypoxic conditions makes them an ideal substance for regulating cell response to environmental changes [34]. Protein hydroxylases, which could oxidize the C-H bond into a C-OH group in an amino acid side chain, belong to 2-oxoglutarate (2OG)/Fe(II) dependent oxygenase family of proteins [35]. The genes encoding 2-oxoglutarate (2OG)/Fe(II) oxygenase detected in aquatic-specific genes of the genus *Methylophilus* can make it achieve hydroxylation process and survive under hypoxic conditions. In addition, biofilms were known to promote the transfer of electrons from cellular to extracellular conditions by binding to self-secreted redox-active compounds such as cytochrome *c* and riboflavin [36,37]. Diguanylate cyclase catalyzes the synthesis of cyclic guanosine monophosphate (c-di-GMP), which plays a key role in the regulation of biofilm formation in bacteria [38]. Matsumoto et al. [39] indicated that the intracellular level of c-di-GMP was sufficient for the expression of extracellular electron transfer (EET)-related genes. We detected genes encoding diguanylate cyclase in aquatic-specific genes, which enabled the genus *Methylophilus* to use alternative electron acceptors instead of oxygen for EET under the hypoxic conditions, so as to survive in the aquatic environment.

We detected genes encoding acyltransferase, which participated in the transport of lipoproteins from the intima to the outer membrane in leaf-specific genes. We speculated that they play an important role in the colonization and survival of the genus *Methylophilus* in leaf habitats. Kovacs-Simon et al. [40] indicated that bacterial lipoproteins have a variety of functions, including membrane synthesis, signal transduction, bacterial adhesion, etc. In general, lipoproteins are transported to the extracellular membrane via the transport system to perform their physiological functions and play an important role in bacterial colonization on the host. Lipoproteins have been found in a variety of bacteria. Roussel-Jazede et al. (2011) and Coutte et al. (2003) found that the *NalP* protein of *Neisseria meningitidis* and *SphB1* protein of *Bordetella pertussis* contain the secretion signal and self-transporter protein structure of lipoprotein [41,42]. They were transported to the outer membrane by the Type V secretory pathway.

### 4.2. The Elemental Metabolism Feature of Methylophilus

In *Methylophilus* genomes, genes including *norB*, *nasA* and *nirB/D* were involved in denitrification, assimilatory nitrate reduction and dissimilatory nitrate reduction, respectively. We speculated that the *Methylophilus* strains reduced nitrate to ammonia by a combination of assimilatory and dissimilatory nitrate reduction and took ammonia as the critical nitrogen source in three ecosystems. While, species in the adjacent genus *Methylotenera* reduced nitrate to nitrogen by denitrification, which differentiates from *Methylophilus* [43]. Microorganisms have great diversity in nitrogen transformation, and each of them has its own optimal growth conditions [44]. In the microbial nitrogen-cycling network, it needs a lot of energy consumption to reduce nitrate to nitrogen by denitrification, as more ATP is in need for electron transfer under a long electron transfer chain [44]. We speculated that *Methylphilus* performed nitrate respiration under hypoxic conditions with less energy consumption via the combination of DNRA and ANRA. Notably, genes encoding nitrite reductase (*nirSK*) were not identified in the nitrogen metabolism of all the 18 genomes. However, seven of the 18 genomes, including strains isolated from leaf and sediment, encoded *norB* genes, which indicates a distinct feature in nitric oxide reduction to nitrous oxide, a notable greenhouse gas. As for the source of nitric oxide, we speculated that it was produced by other microorganisms co-occurring in the methanotrophic consortia.

In this study, a distinct way in methanol metabolism was detected to convert trimethylamine N-oxide to formaldehyde in strains from the sedimentary habitats. Trimethylamine N-oxide can participate in microbial respiration as an alternative electron acceptor under anaerobic conditions and then be reduced to formaldehyde for further metabolism [28]. This provided some hints for the survival mechanism of the genus *Methylphilus* in the anaerobic sedimentary environment. In addition, genes encoding enzymes that catalyzed the conversion of sulfate to sulfur were identified in *Methylophilus* genomes, indicating that strains might also use sulfate as an alternative electron acceptor under the oxygen-depleted condition. Moreover, many genes existed in clusters in the genomes, and the integrity of the clusters affected methanol, nitrogen and sulfur metabolism. For example, the *mxaACKLD-mxaIGJ* operon, including eight methanol oxidation genes, was present in the *Methylophilus* genomes. Notably, besides the *MxaI* gene encoding beta subunit of MDH, we only found a gene containing a PQQ-like domain adjacent to the cytochrome *c_L_* (*mxaG*) gene in *Methylophilus* genomes, which was poorly characterized. We speculated that it might be the synonym of the *MoxF/MxaF* gene encoding the alpha subunit of MDH to complete the process of methanol oxidation. 

In summary, comparative genomics can identify coded, non-coded, and endemic sequences by comparing the genomes of different related species, which is advantageous for genomic analysis; specifically, both intraspecific and interspecific genomic comparisons can be carried out to provide the foundation for future research on other methylotrophic bacteria, such as *Methylotenera*, *Methylovorus* and *Methylobacillus*. According to the analysis, *Methylophilus* can use single-carbon compounds (methanol) as a carbon and energy source, and drive sustainable and clean utilization of it, such as building cell factories as hosts and synthesizing amino acids and cell proteins. In addition, artificially constructed methylotrophs can improve methanol utilization efficiency and promote methanol bioconversion. Thus it is of great practical significance to study the metabolism of *Methylophilus*. With the completion of the whole-genome sequencing of increasing *Methylophilus* strains, bioinformatics, transcriptome and proteomics can also be used to study their metabolic pathways using genomic information [45]. The complete genomic sequences provided the possibility of further research of the genus *Methylophilus* and other related methylotrophs.

## 5. Conclusions

In this study, the genus *Methylophilus* was investigated using comparative genomic analysis, and three newly isolated strains were obtained using the enrichment culture. The analysis indicated that there were differences among strains from different environments (leaf, water and sediment). In addition, the *Methylophilus* genomes consisted of gene inventories related to the metabolism of nitrogen, methanol and sulfur and cellular electron transfer mechanism. However, in the genus *Methylophilus,* part of the genes responsible for metabolism was absent, thus showing that it had metabolic restrictions. Thus, the functional genes of isolated strains deserve more attention to identify the specific metabolic process. To sum up, this study enhanced our understanding of the fundamental genomic features of the genus *Methylophilus* and the impact of heterogeneous environment on the genus *Methylophilus* at the genomic level, revealed the mutual adaptation between the genus *Methylophilus* and different environments and the potential survival mechanism of it under anaerobic conditions, providing insights into methylotrophic bacteria.

## Figures and Tables

**Figure 1 microorganisms-09-01577-f001:**
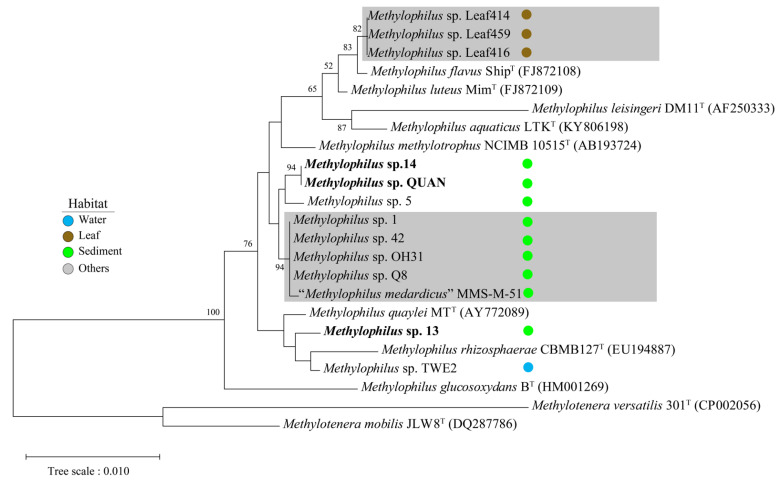
Phylogenomic tree based on 16S rRNA genes. The positions of the 13 strains of genus *Methylophilus*, 8 *Methylophilus* type strains and 2 *Methylotenera* strains were shown in the phylogenomic tree and bootstrap values (the percentage of 1000 data resamplings) ≥50% are shown at the nodes based on neighbor-joining method. Three newly isolated methylotrophic strains (*Methylophilus* sp. 13, 14 and QUAN) were highlighted with bold. The 16S rRNA gene sequences of *Methylotenera versatilis* 301^T^ and *Methylotenera mobilis* JLW8^T^ were used as the outgroup. The scale bar represents one substitution per 100 bp. The clades belonging to the same species were highlighted with grey according to AAI results (AAI values larger than 95%), respectively. The solid circles with colors, blue, brown, green and grey, represent isolation sources, water, leaf, sediment and others, respectively. 8 *Methylophilus* type strains and 2 *Methylotenera* strains were unlabeled habitat sources. 2 *Methylophilus* type strains, *Methylophilus rhizosphaerae* and *Methylophilus methylotrophus* harbored whole-genome sequences.

**Figure 2 microorganisms-09-01577-f002:**
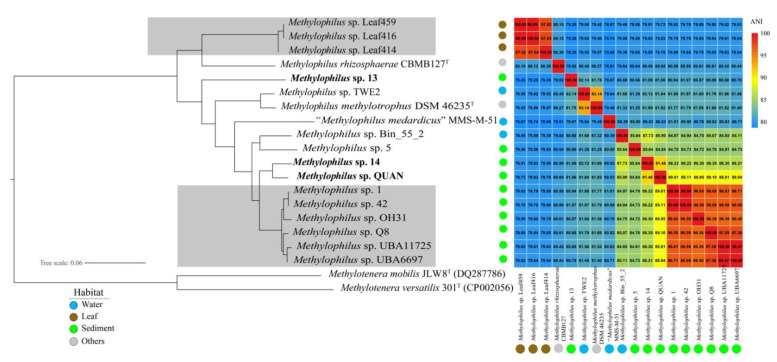
Phylogenomic tree based on 1035 single-copy genes with *Methylotenera versatilis* 301^T^ and *Methylotenera mobilis* JLW8^T^ as the outgroup and ANI heatmap. The positions of the 18 strains of genus *Methylophilus* and 2 type strains of genus *Methylotenera* were shown in the phylogenomic tree based on the comparative sequence analysis (1,035 single-copy genes). Three newly isolated methylotrophic strains (*Methylophilus* sp. 13, 14 and QUAN) were highlighted with bold. The clades belonging to the same species were highlighted with grey according to ANI results (ANI values larger than 95%), respectively. The solid circles with colors, blue, brown, green and grey, represent isolation sources, water, leaf, sediment and others, respectively. The tree scale (0.06) indicated the number of substitutions per amino acid position. Cells in the heatmap consistent with different ANI values were represented by different color scales.

**Figure 3 microorganisms-09-01577-f003:**
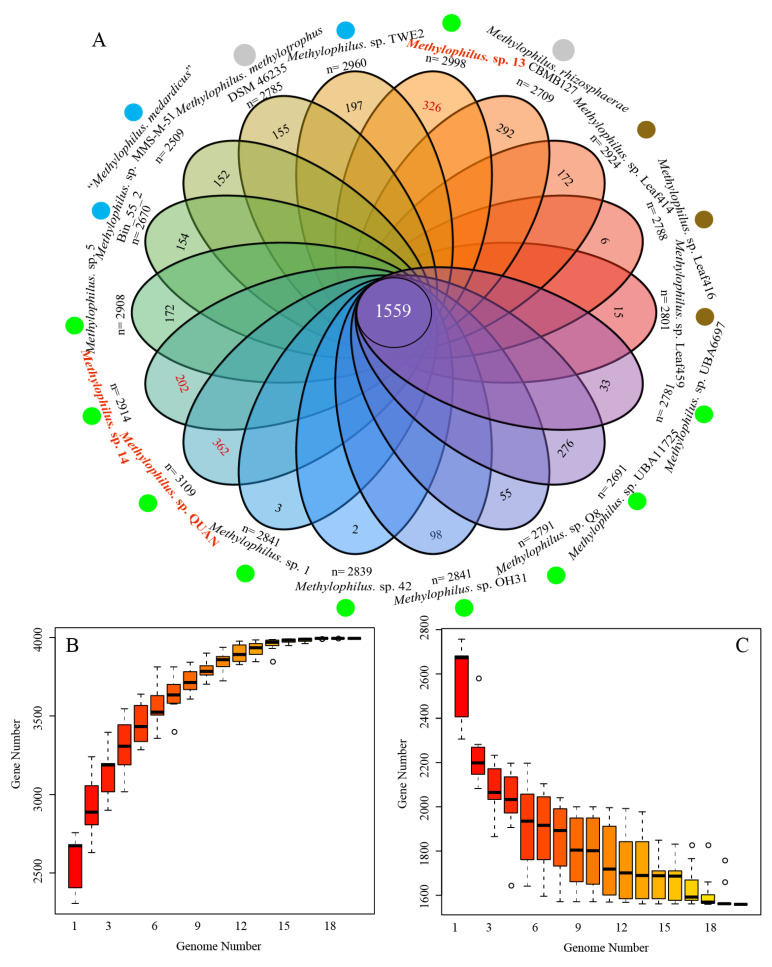
(**A**) Flower petal plot of 18 *Methylophilus* genomes. The numbers of yielded unique genes of each strain were shown on the flower petal plot. The center circle indicated the number of core genes obtained from 18 strains. The number below the strain name represented the gene number. The solid circles with colors, blue, brown, green and grey, represent isolation sources, water, leaf, sediment and others, respectively. (**B**) Pan-genome plot of genus *Methylophilus* members. It reached a maximum number of 3994 genes. (**C**) Core-genome plot of genus *Methylophilus* members. The core genome size is 1559 genes.

**Figure 4 microorganisms-09-01577-f004:**
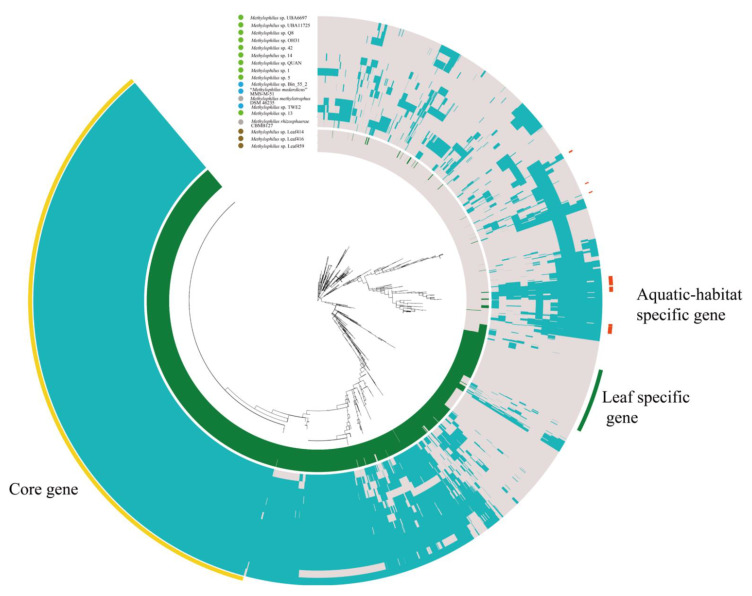
Orthology analysis. Clustering of genomes based on the presence/absence patterns of 3994 orthologous clusters. Genomes of strains isolated from leaf and aquatic habitats (water and sediment) were colored with green and blue in the gene heatmap. Eighteen genomes were defined by the gene tree in the center, and they were organized in radial layers as aquatic-habitat specific gene, leaf specific gene and core gene, which were colored with red, green and yellow. The solid circles with colors, blue, brown, green and grey, represent isolation sources, water, leaf, sediment and others, respectively.

**Figure 5 microorganisms-09-01577-f005:**
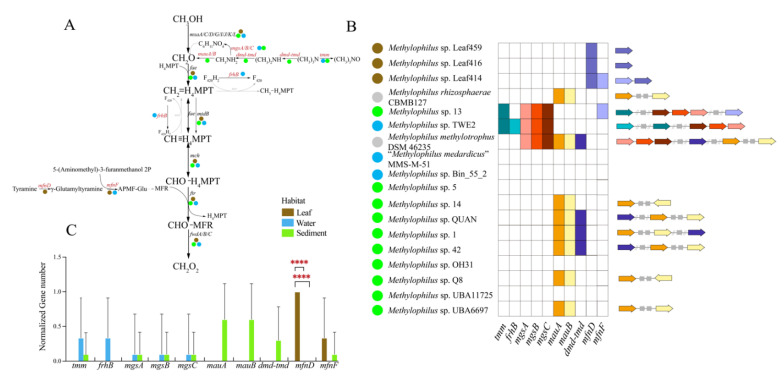
(**A**) Methanol metabolic pathway. Each step of the reaction is tagged with the corresponding genes. Genes that are present in all genomes are highlighted in black, while those that are present in partial genomes and absent in all of the genomes are highlighted in red and grey, respectively. (**B**) Distribution of partial methanol metabolism genes in *Methylophilus* genomes. The colored boxes indicate that the gene is present in the genome, whereas the white box indicates that it is absent. A functional gene clusters structural plot was performed according to the heatmap. The colored arrows correspond to the colored boxes. The grey arrows represent unknown genes. The solid circles with colors, blue, brown, green and grey, represent isolation sources, water, leaf, sediment and others, respectively. (**C**) Histogram constructed by computing the normalized number of partial methanol metabolism genes involved in (**B**). Three box colors represented three environments. Significant differences in *mfnD* gene between leaf versus water and leaf versus sediment by one-way ANOVA. ****: *p* < 0.00001.

**Figure 6 microorganisms-09-01577-f006:**
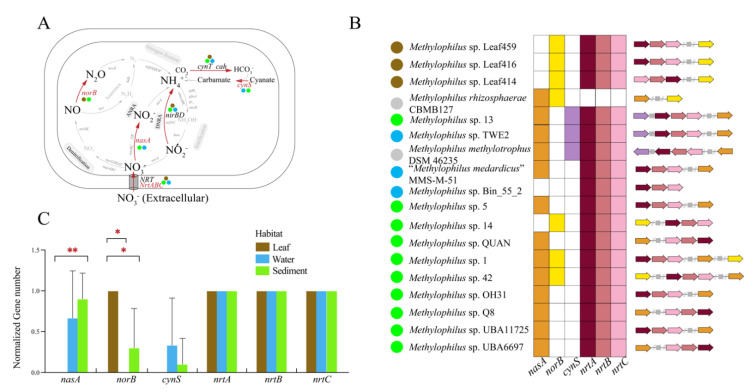
(**A**) Nitrogen metabolic pathway. Each step of the reaction is tagged with the corresponding genes. Genes that are present in all genomes are highlighted in black, while those that are present in partial genomes and absent in all of the genomes are highlighted in red and grey, respectively. (**B**) Distribution of partial nitrogen metabolism genes in *Methylophilus* genomes. The colored boxes indicate that the gene is present in the genome, whereas the white box indicates that it is absent. Functional gene clusters structural plot was performed according to the heatmap. The colored arrows correspond to the colored boxes. The grey arrows represent unknown genes. The solid circles with colors, blue, brown, green and grey, represent isolation sources, water, leaf, sediment and others, respectively. (**C**) Histogram constructed by computing the normalized number of partial nitrogen metabolism genes involved in (**B**). Three box colors represented three environments. Significant differences in *nasA* gene between leaf versus sediment and in *norB* gene between leaf versus sediment and water versus sediment by One-Way ANOVA. **: *p* < 0.01; *: *p* < 0.05.

**Figure 7 microorganisms-09-01577-f007:**
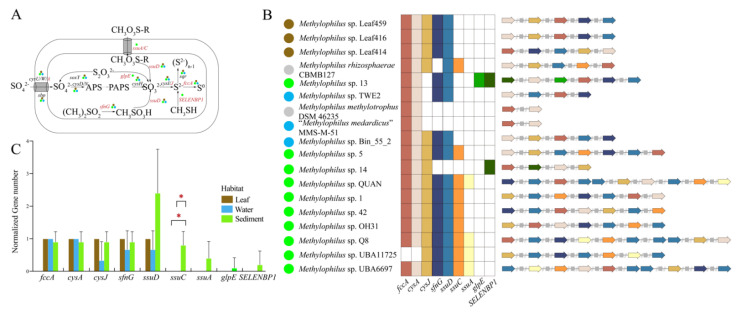
(**A**) Sulfur metabolic pathway. Each step of the reaction is tagged with the corresponding genes. Genes that are present in all genomes are highlighted in black, while those that are present in partial genomes are highlighted in red. (**B**) Distribution of partial sulfur metabolism genes in *Methylophilus* genomes. The colored boxes indicate that the gene is present in the genome, whereas the white box indicate that is absent. Functional gene clusters structural plot was performed according to the heatmap. The colored arrows correspond to the colored boxes. The grey arrows represent unknown genes. The solid circles with colors, blue, brown, green and grey, represent isolation sources, water, leaf, sediment and others, respectively. (**C**) Histogram constructed by computing the normalized number of partial sulfur metabolism genes involved in (**B**). Three box colors represented three environments. Significant differences in *ssuC* gene between leaf versus sediment and water versus sediment by one-way ANOVA. *: *p* < 0.05.

**Figure 8 microorganisms-09-01577-f008:**
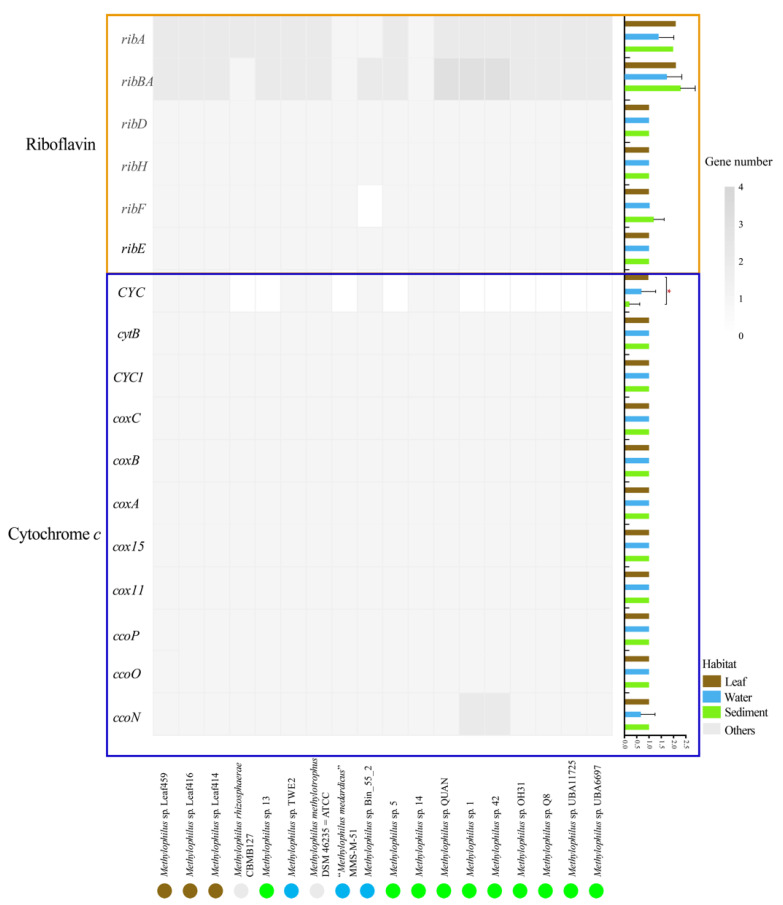
Distribution of genes encoding cytochrome *c* and riboflavin in *Methylophilus* genomes. The colored boxes indicate that the gene is present in the genome, whereas the white box indicates that it is absent. The solid circles with colors, blue, brown, green and grey, represent isolation sources, water, leaf, sediment and others, respectively. The histogram was constructed by computing the normalized number of all genes encoding cytochrome *c* and riboflavin. Three box colors represented three environments. The boxplots were constructed by computing all gene numbers. Three box colors represented three environments. Significant differences in *CYC* gene between leaf versus sediment environment by one-way ANOVA. *: *p* < 0.05.

**Table 1 microorganisms-09-01577-t001:** General features of the genomes within the genus *Methylophilus*.

Stains	Taxonomy	Genome Assembly	Genome Size (Mb)	ScafNum	G+C Content (%)	Gene Number	16S rRNA	tRNAs	Completeness	Contamination	Source Habitats
Leaf459	*M. flavus*	GCA_001425495.1	2.95	16	48.29	2801	5	30	99.57	0	Arabidopsis leaf
Leaf416	*M. flavus*	GCA_001424665.1	2.94	8	48.25	2788	1	40	100	0	Arabidopsis leaf
Leaf414	*M. flavus*	GCA_001425425.1	3.03	9	48.24	2924	1	44	100	0	Arabidopsis leaf
CBMB127 *	*M. rhizosphaerae*	GCA_900100975.1	2.76	10	51.35	2709	1	41	100	0.03	Rhizosphere soil of rice
13	*M.* *quaylei*	GCA_015354335.1	3.11	14	51.13	2998	1	41	100	0	Lake Fuxian Sediment
TWE2	*M. quaylei*	GCA_001183865.1	3.08	1	49.54	2960	2	40	100	0	Tap water incubated with methane
DSM 46235 *	*M. methylotrophus*	GCA_000378225.1	2.86	14	49.61	2785	1	40	100	0	Activated sludge
MMS-M-51 *	“*M. medardicus*”	GCA_006363915.1	2.60	1	49.82	2509	2	38	100	0	Water column of Lake Medard, Czechia
Bin_55_2	Inconclusive	GCA_008015755.1	2.75	40	50.49	2670	0	41	95.41	1.28	Advanced Water Purification Facility
5	*M. methylotrophus*	GCA_000515275.1	3.03	1	50.25	2908	3	40	100	0.11	Lake Washington Sediment
14	*M. methylotrophus*	GCA_015354345.1	3.02	15	50.48	2914	1	40	100	0.43	Lake Fuxian Sediment
QUAN	*M. methylotrophus*	GCA_015354445.1	3.15	6	50.33	3109	1	42	100	0.85	Lake Fuxian Sediment
1	*M. methylotrophus*	GCA_000374225.1	2.97	1	50.43	2841	2	40	100	0	Lake Washington Sediment
42	*M. methylotrophus*	GCA_000384155.1	2.97	1	50.44	2839	1	44	100	0	Lake Washington Sediment
OH31	*M. methylotrophus*	GCA_000576615.1	2.93	16	50.59	2841	1	46	100	0	Pond Sediment in Hokkaido, Japan
Q8	*M. methylotrophu* *s*	GCA_000800115.1	2.90	1	50.63	2791	3	42	100	0.03	Lake Washington Sediment
UBA11725	Inconclusive	GCA_003538435.1	2.22	303	50.75	2691	0	42	78.45	0	Freshwater sediment
UBA6697	Inconclusive	GCA_002454655.1	2.82	51	50.64	2781	0	25	97.93	0	Costa Rican marine sediment

* Type species.

## Data Availability

The genomic sequences of three strains (sp. 13, 14 and QUAN) in this study were submitted to the GenBank database under accession numbers GCA_015354335.1, GCA_015354345.1, and GCA_015354445.1, respectively.

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
