# Peer review of "Comparative Genomics Revealing Insights into Niche Separation of the Genus Methylophilus"

_microorganisms, 2021, doi:10.3390/microorganisms9081577_

Round 1
Reviewer 1 Report
Major comment:
Herein, a manuscript describing the sequencing of the genus Methylphilus isolate is reported. The manuscript could, however, significantly improve with a better presentation and discussion of the results:
Major points need to be addressed, and in particular, no functional genomics was provided. In my opinion, evaluate at least the expression of selected genes may represent an interesting point of view in the light of the role of such genus in nutrient cycling.
This script needs to put the discussion into an ecological context. To me, the question remains, why studying this bacterium is relevant. Is it particularly abundant, or could it play an essential role within the respective microbial community? I am missing the bigger picture and an ecological hypothesis in the introduction (which is overall very short) and also in the discussion. The new results are also not really set into perspective with the other Methylphilus genomes analyzed. What is the ecological question? What conclusions do the authors draw? Why does it make sense to compare these genomes from such different environments? Even, there are few additional genomes (n=15). Please check other MAGs.
Specific comments:
L13, replace to three strains (13, 14, and QUAN) related to the genus Methylophilus
L16, insert “, respectively.”
L17-18, there are 1422 genes identified as single-copied and core across 18 strains? Cuz, for phylogenomic analysis, we have to compare those genes between strains.
L19, probably, the gene contents and physiological characteristics for some microbugs are completely different between two environments.
L20, high similar? Please provide the ranges.
L21-25, rewrite. Already genomics level, these stains might be similar in the same environments. How were your strains in genomic level? Which genes were highlighted or reduced among comparative genomic analyses? Please highlight your main result and discussion and theoretical hypothesis.
L33, well, it’s too difficult to say autotrophic and heterotrophic microorganisms in the methylotrophs. There are obligate methanotrophs and methylotrophs based on the used E source methane or not (other C1 compounds even methylamine).
L36, land indicated for what? WWTP? Wetland?
L39, yes some members of the genus Bacillus and other genera can use C1 compounds during cell growth. However, we could not say they are or might methylotrophs. Should be re-considered.
L40-44, if you wanted to say C1 bioconversion, please rewrite. Many organisms can use methanol as a carbon source. Also, methanotrophs can be only producing methanol during methane oxidation.
L45-47, should be combined into a single paragraph with L30-56. And more focused.
L59, however, in this study, you did not investigate other methylotrophs. Right?
L72, the genus Mehtylophilus.
L74, which were the differences between genome-level and genome-encoded traits?
L77, other fifteen strains belong to the genus Methylophilus, it might be more suitable.
L89-113, it’s confusing. Why did you use methane for enrichment culture at 18C during only one week? And then, the authors spent two weeks, to isolate these organisms on the agar plate. How did you provide methanol during agar plate culture? During agar plate preparation, how did you keep methanol evaporations? Also, correct agar 1.5~2.0 %(w/v). 0.5% (v/v) methanol.
In addition, provide more details for isolation and 16S rRNA gene analysis.
L130, remove sp.
L153-160, in fact, it’s not comparative genome analysis.
L163-164, it’s prepared by Chinese. So, how do we access and can understand? Even. Section 2.5 was not a metabolic analysis. Statistics were included.
L170-172, please provide the coverage.
Table1. please define the taxonomy for inconclusive genomes using 16S rRNA gene. Also, it’s curious results. Incomplete genomes (e.g scaf num) were completeness 100%? Based on the table 1, most genome are not completed.
L181-182, it seemed not completed sentence. Phylogenetic.. figure 1 and figure 2) …
Figure 1 and 2, please re-construct the trees using obvious outgroup.
L188-192, have to analyze ANI, AAI or GGDC etc. then, we could say novel species candidate or not. Using only one gene (i.e 16S rRNA gene), we cannot say that in the genomic era.
L203 (figure 2), change to “isolation source” or “habitat”
L209-252, long story.. and detouring. There were only 18 genomes. Using these tiny genomes, it’s difficult to say any biostatistical results. Please reduce the sentence.
L254, well, it might be correct. “the Methylophilus genomes selected in this study”
L278-280, please provide any scientific reasons that these genes might be related to aquatic-specific genes. Your genomes contained these genes (n=23)?
L290-299, this section should be moved into 3.3
L273-317, in this part, the authors tried to interpret any genes related to their habitats. However, there was only a simple gene list. So, please have to find any scientific clue for your hypothesis.
L319, please check enzyme (capital letter) or gene (italic)
L411, extracellular electron transfer analysis? But not cellular electron transfer for respiration??
L321-445, in this long part, the authors deduced something based on the existence for some gene or not. It seemed to make sense.
L448, sorry, the authors just selected only 15 genomes from GenBank. Did you check GTDB-Tk?
L462-476, in this part, how can we infer to any ecological advantages for methylotrophs against other competitors?
L478-490, did you check the activity for denitrification or DNAR using your isolates?
L484, who are there in the family Methylophilaceae?
L488-492, it confused. “were not present” indicated in your genomes?
L493-494, provide the full naming for H4MPT and RUMP.
L495-497, there is sulfur limitation environments? I meant their habitats. It might be possible that sulfate converts to sulfur under anaerobic conditions.
L499, define “elemental”
L499-507, your isolates and other microorganisms (classified into the genus Methylophilus) are methylotrophs. Therefore, they have a capability for methanol oxidation. Also, did you check maxFI genes encoding alpha and beta subunit of MDH in your genomes?
L508-516, if they could not produce riboflavin in their habitats, so what is your speculation? As a minor-vital component, vitamins are essential for their growth. How can survive w/o vitamin support?
L517-531, please see the major comments. You need to analyze physiological characterization.
L532, after reorganization, reanalysis, it will be changed in the revised script, hopely.
Two supplementary figures: please provide a more detailed legend.
Reviewer 2 Report
this study is devoted to the comparative analysis of three isolated methylotrophic bacteria with known representatives of genus Methylophilus, including on the basis of available genome-wide sequences. In my opinion, the work is very interesting, corresponds to the profile of the journal Microorganisms and can be published after minor corrections.
- the materials and methods should indicate where the samples were taken for the isolation of microorganisms. This information is in the Abstract, but it must also be represented in the methods.
- throughout the text, the authors should italicize genes and not only generic, but also specific names of microorganisms (for example, lines 188, 244, 269, 370, 374, 375, 425, 426, 568, 576, 578603, 607, 611 etc.
- for cytochrome c everywhere in the text "c" should be made in italics
- Line 269 (Fig 3) “genoomes” – what is this?
- Line 92. the expression "culture was extracted", in my opinion, requires paraphrasing, since the cultures were isolated by sieving on an agarized medium, and extraction involves a chemical process
- Line 536 you don't need a comma before the parenthesis
Reviewer 3 Report
Journal: Microorganisms
Title: Comparative genomics revealing insights into niche separation of the genus Methylophilus
Article ID: Microorganisms-1249801
In this manuscript, a comparison among genomics of bacterial strains of te genus Methylophilus was carried out. Insights about niches separation of the genus were highligted, and , 3 Methylophilus strains (sp. 13, 14 and 13 QUAN), newly isolated from Lake Fuxian, were compared with 15 other sequenced strains within 14 the genus Methylophilus at the genomic level.
The analysis indicated that there were significant differences among strains from leaf, water and sediment. In addition, the Methylophilus genomes consisted of gene inventories related to the metabolism of nitrogen, methanol and sulfur and extracellular electron transfer (EET). This study enhanced knowledge of the fundamental genomic features of the genus Methylophilus, revealed the differences among diverse environments and providing insights into methylotrophic bacteria.
Concerning Supplementary Materials, it is not possible to obtain this material online at www.mdpi.com/xxx/s1. Figure S1: AAI heatmap, and Figure S2: Heatmap conducted according to CAZy results, are available for consultation, only.
Table S1: 16S rRNA gene sequence similarity and ANI values, Table S2: Specific genes of 18 genomes among various environments, Table S3: Statistical analysis data in methanol metabolism, nitrogen metabolism, sulfur metabolism and EET mechanism, are not available.
The manuscript is interesting, results are important, anyhow results are reported in a not clear way in different points.
Revisions
Lines 32-34: ‘These bacteria are divided into facultative and obligate methylotrophs (type I, type II and type X methylotrophs) and autotrophic and heterotrophic methylotrophs.’ the sentence is not clear, please xplain better;
line 72: ‘three novel species of Methylophilus’ change to ‘three novel species of the genus Methylophilus’;
line 109: ‘Methylotenera’ change to Italic style;
Line 179: Table 1: Change the species names to Italic style;
line 186: ‘Supplementary Table 1’: this file is not present in the supplementary material, as Supplementary Figure 1 and Supplementary Figure 2 are present, only;
line 190: ‘Methylotrophus’ change to Italic style;
Line 202: Figure 2: Numbers in the squares of the right part of the Figure is difficult to read, although colors discriminates among similarities;
line 236: ‘M.’ change to the name of the genus in extenso ‘Methylophilus’;
line 238: please, include references;
lines 243-244: ‘Methylophilus genomes … M. rhizosphaerae’ change to Italic style;
line 269: Figure 3A: the nomenclature is not correct, as the name of the genus should be reported in extenso. Again, the same name should be written in Italics. Please, avoid to use the Italic style to distinguisch the three news isolated strains … maybe the different color is enough or the bold charachter ….;
line 269: ‘genoomes’ change to genomes;
line 280: ‘cis-trans’ chnge to Italic style;
line 281: ‘Supplementary Table 2’: this file is not present in the supplementary material, as Supplementary Figure 1 and Supplementary Figure 2 are present, only;
line 319: in paragraph 3.3 change the genes names in Italic style;
line 322: Figure 5A is cited in the text, not the other parts of Figure 5 are reported;
line 351: (Fig.5): ‘mfnD’ change to Italic style and the names of genes along with the manuscript;
line 354: ‘norB’ change to Italic style and the same along with the text of the manuscript;
lines from 342 and from 388: change the names of genes in Italic style;
line 388: In the paragraph, Figure 7A and Figure 7C are cited, Figure 7B was not described;
lines 390-393: this period is not clear, it should be reported that the subjects are the genes, to be reported in Italic style. At line 393: ‘… monooxygenase sfnG were detected …’ change to ‘… monooxygenase sfnG genes were detected …’;
line 411: from here and in the following: ‘citocrome c’ change ‘c’ to Italic style;
Figg. 5, 6, 7, and 8: ‘Methylophylus’ change to Italic style;
line 440: ‘citocrome c’, change ‘c’ to Italic style;
lines 523-524: please, explain better this concept …’ … to solving the problem of environmental pollution …’ as the compounds with a single C atom are very difficult to degrade …;
lines 548-549: Was Table S3: (Statistical analysis data in methanol metabolism, nitrogen metabolism, sulfur metabolism and EET mechanism.) Reported in the text of the manuscript?;
line 563: References: The titles of articles and chapters must not be reported in capital letters, and all the species names should be reported in Italic style.
Round 2
Reviewer 1 Report
congratulations. you have addressed all comments. now, your manuscript will be estimated by readers.
Author Response
We really appreciated the comments and suggestions from you. It’s our luck to meet you by the journal Microorganisms. Also, we are pleased that our efforts have paid off, and this article finally has a chance to meet readers.
Reviewer 3 Report
All recommendations that emerged during the previous review were considered by the authors.
Only, please, if the authors can consider changing the gene markers in Tab. S3 to Italic style.
Author Response
The gene markers in Table S3 were changed to italic style.